# Significance of Hypermethylation of Tumor-Suppressor Genes PTGER4 and ZNF43 at CpG Sites in the Prognosis of Colorectal Cancer

**DOI:** 10.3390/ijms231810225

**Published:** 2022-09-06

**Authors:** Chao-Yang Chen, Jia-Jheng Wu, Yu-Jyun Lin, Chih-Hsiung Hsu, Je-Ming Hu, Pi-Kai Chang, Chien-An Sun, Tsan Yang, Jing-Quan Su, Yu-Ching Chou

**Affiliations:** 1Division of Colorectal Surgery, Department of Surgery, Tri-Service General Hospital, National Defense Medical Center, Taipei 114, Taiwan; 2School of Public Health, National Defense Medical Center, Taipei 114, Taiwan; 3Graduate Institute of Medical Sciences, National Defense Medical Center, Taipei 114, Taiwan; 4Department of Public Health, College of Medicine, Fu-Jen Catholic University, New Taipei City 242, Taiwan; 5Data Science Center, College of Medicine, Fu-Jen Catholic University, New Taipei City 242, Taiwan; 6Department of Health Business Administration, Meiho University, Pingtung 912, Taiwan; 7Department of Medical Education and Research, Kaohsiung Veterans General Hospital, Kaohsiung 813, Taiwan

**Keywords:** colorectal cancer, PTGER4, ZNF43, CpG island hypermethylation, tumor tissue, adjacent normal tissue, prognostication

## Abstract

The status of DNA methylation in primary tumor tissue and adjacent tumor-free tissue is associated with the occurrence of aggressive colorectal cancer (CRC) and can aid personalized cancer treatments at early stages. Tumor tissue and matched adjacent nontumorous tissue were extracted from 208 patients with CRC, and the correlation between the methylation levels of *PTGER4* and *ZNF43* at certain CpG loci and the prognostic factors of CRC was determined using the MassARRAY System testing platform. The Wilcoxon signed-rank test, a Chi-square test, and McNemar’s test were used for group comparisons, and Kaplan–Meier curves and a log-rank test were used for prediction. The hypermethylation of *PTGER4* at the CpG_4, CpG_5, CpG_15, and CpG_17 tumor tissue sites was strongly correlated with shorter recurrence-free survival (RFS), progression-free survival (PFS), and overall survival (OS) [hazard ratio (HR) = 3.26, 95% confidence interval (CI) = 1.38–7.73 for RFS, HR = 2.35 and 95% CI = 1.17–4.71 for PFS, HR = 4.32 and 95% CI = 1.8–10.5 for OS]. By contrast, RFS and PFS were significantly longer in the case of increased methylation of *ZNF43* at the CpG_5 site of normal tissue [HR = 2.33, 95% CI = 1.07–5.08 for RFS, HR = 2.42 and 95% CI = 1.19–4.91 for PFS]. Aberrant methylation at specific CpG sites indicates tissue with aggressive behavior. Therefore, the differential methylation of *PTGER4* and *ZNF43* at specific loci can be employed for the prognosis of patients with CRC.

## 1. Introduction

Colorectal cancer (CRC) was the third most occurring malignancy and the second main cause of cancer-related mortality globally in 2020 [1]. More than 1.9 million new cases of CRC and approximately 0.9 million CRC-related deaths were reported in 2020 [2]. The global incidence of CRC is predicted to increase by 80% by 2035, with an estimated 2.4 million newly diagnosed cases and 1.3 million CRC-related deaths around the world [3]. Cancer staging is essential for determining the extent of CRC invasion after a confirmed diagnosis; information on cancer staging is foundational to an accurate prognosis and appropriate choice of treatment. The tumor-node-metastasis (TNM) staging system (8th edition, 2017), established by the American Joint Committee on Cancer (AJCC) and the Union for International Cancer Control (UICC), is extensively used for CRC evaluation worldwide [4]. However, various survival patterns and treatment responses have been observed among patients within the same clinicopathologic stage because of the heterogeneity of colorectal neoplasms [5]. The survival rate of patients with CRC is low, especially among those with postoperative recurrence or progression even after tumor resection, in spite of recent developments in multidisciplinary techniques, such as those involving surgery, chemotherapy, radiotherapy, targeted therapy, and immunotherapy [6]. Methylation of the *MGMT* and *IFBP3* genes and the CpG island methylator’s phenotype status could be used to predict the treatment effect of 5-fluorouracil-based chemotherapy [7]. The data of the KEYNOTE-177 trial reinforced that the anti-PD-1 antibody pembrolizumab demonstrated a long-lasting antitumor effect to improve progression-free survival and a lower incidence of adverse reactions than chemotherapy, which was regarded as a first-line therapy in metastatic CRC patients with DNA mismatch repair-deficiency or microsatellite instability-high [8]. Regorafenib, a multi-functional kinase inhibitor that targets cell proliferation, signals transduction and tumor angiogenesis, showed clinical benefits in metastatic CRC with administration of an appropriate dosage [9]. A comprehensive postoperative examination is recommended for patients with CRC who undergo radical surgery for the timely detection of early-stage recurrence, especially in patients without any symptoms; this is key to ensuring that treatment is administered as soon as possible [10]. Therefore, relevant biomarkers should be identified to help oncologists discern patient prognoses and monitor cancer recurrence in different clinical scenarios.

Studies conducted over the past few decades have demonstrated that epigenetic alterations are potential clinically significant molecular markers of CRC; this is because epigenetic alterations are involved in the pathophysiological pathways of carcinogenesis in the early stages of oncogenesis [11]. Furthermore, these alterations can be utilized to enhance the identification of patients with high risks of early recurrence, distant metastasis, and death [12]. Hypermethylation is a key indicator of CRC and occurs at CpG dinucleotides in the promoter regions of genes that regulate diverse cell functions, such as signal transduction, cell cycle, apoptosis, DNA repair, and transcription-suppressing metabolism [13,14]. Therefore, increased methylation at certain CpG sites of genes can aid in the prognosis of CRC. 

*PTGER4* is a major *PGE2* receptor and is involved in the feed-forward pathway that upregulates *STAT3* phosphorylation and activates *PTGS2* expression in the tumor cells of CRC [15]. *ZNF43* is a part of the Krüppel-associated box domain zinc finger (*KRAB-ZNF*) gene family, and its expression considerably differs across tissues. A few members of the *KRAB-ZNF* family behave as epigenetic suppressors that promote DNA methylation, and repressive chromatin marks are defined accordingly [16]. In this study, we investigated the status and extent of DNA methylation in neoplastic samples and in paired adjacent normal-appearing samples to determine the relationship between the methylation signature of candidate genes and the 5-year prognosis of CRC. The tissue-specific DNA methylation of *PTGER4* and *ZNF43* facilitates the prognosis of CRC and the provision of timely intervention.

## 2. Results

### 2.1. Clinical Features and Follow-Up Information of Patients

A total of 208 patients with CRC who underwent surgical treatment at TSGH were enrolled into the study. Tumor tissue and adjacent normal tissue were extracted for analysis. The correlation between the methylation status of target genes and the demographic and clinical variables of patients with CRC can be inferred from the data summarized in Table 1. The 5-year OS rate was considerably higher in patients with *PTGER4*-methylated normal tissue than in the patients without *PTGER4*-methylated tissue. The patients with *ZNF43*-methylated normal tissue were at earlier cancer stages than those without *ZNF43*-methylated tissue. No other correlation was observed between the methylation statuses of the selected genes and the clinical features of the participants.

### 2.2. Correlation between PTGER4 and ZNF43 Methylation Status and CRC

The proportion of methylated genes was higher in neoplastic samples than in adjacent non-neoplastic samples (*PTGER4*, 61.1% vs. 45.1%; *ZNF43*, 73.6% vs. 29.2%); statistical significance was determined using McNemar’s test (*PTGER4*, *p* = 0.017; *ZNF43*, *p* < 0.001) (Table 2). We determined the correlations of the methylation statuses of *PTGER4* and ZNF43 with the 5-year RFS, PFS, and OS of the patients. The univariate analysis results for *PTGER4* methylation indicated that patients diagnosed with Stage III + IV CRC exhibited unfavorable RFS [HR (95% CI) = 6.32 (3.11–12.86)], which remained significant even after confounding variables were included in the multivariable analysis for normal tissue and tumor tissue [HR (95% CI) = 6.67 (2.31–19.28) for normal tissue; 6.70 (2.31–19.41) for tumor tissue]. With regard to *ZNF43* methylation, no correlation was noted between the methylation status of *ZNF43* and the likelihood of CRC. However, the Kaplan–Meier survival curves exhibited a different trend, indicating that the methylation of *ZNF43* may have influenced the RFS, PFS, and OS of the patients.

### 2.3. Methylation Level at Certain CpG Sites in Candidate Genes and Its Influence 

To analyze the patterns of gene methylation in patients with CRC, we evaluated the methylation levels of *PTGER4* and *ZNF43* in both tumor tissue and adjacent normal tissue. Higher methylation levels of *PTGER4* were observed in the adjacent tumor-free area than in the primary tumor tissue at the following loci: CpG_3, CpG_4.5, CpG_6.7, CpG_9.10, CpG_11, CpG_13, CpG_15, CpG_16, and CpG_18. By contrast, a larger proportion of *ZNF43* methylation was located in the CRC tissue than in the adjacent normal tissue at the following loci: CpG_2, CpG_3, CpG_4, CpG_5, CpG_6, and CpG_7.8. The *p* values corresponding to these results were <0.05. Figure 1 displays all of the CpG sites of *PTGER4* and *ZNF43* methylation that were not fully detected due to the primer design for sequencing and the sensitivity of the mass spectrometer (Table 3).

To achieve a desired level of statistical power, the methylation level of the target genes were categorized into hypermethylation and hypomethylation groups, and the cut-off set for dichotomization was defined as the median value, as displayed in Table 3. Thereafter, the correlation of the *PTGER4* and *ZNF43* methylation levels at certain CpG sites with clinical outcomes was evaluated. With regard to *ZNF43* methylation, CpG_5 hypermethylation in normal tissue corresponded to a considerably shorter RFS and PFS than CpG_5 hypomethylation, as determined through a Cox multivariate analysis [aHR (95% CI) = 2.33 (1.07–5.08) for RFS and 2.42 (1.19–4.91) for PFS] (Table 4).

Hypermethylation in approximately half of all the CpG sites of tumor tissue that underwent *PTGER4* methylation exhibited an unfavorable 5-year RFS, PFS, and OS, as indicated by the results of the Cox regression analysis (Table 4) and the Kaplan–Meier survival analysis. Therefore, CpG_4.5, CpG_15, and CpG_17 were selected because the patients who underwent hypermethylation at these CpG sites corresponded to undesirable clinical outcomes, as indicated by the results of a Cox regression (Table 4) and separation of the Kaplan–Meier survival curves. Further, the correlation between the hypermethylation of the diverse combinations of the aforementioned CpG sites and patient outcomes (Table 5) was assessed. All methylation levels at the CpG_4.5, CpG_15, and CpG_17 loci of *PTGER4* methylation in neoplastic tissue above the site-specific optimum threshold values were correlated with significantly worse outcomes after adjustment for related confounding factors [aHR (95% CI) = 3.26 (1.38–7.73) for RFS, 2.35 (1.17–4.71) for PFS, and 4.32 (1.8–10.5) for OS]. The results of the Kaplan–Meier analysis of 5-year RFS, PFS, and OS indicated significant differences between the groups with versus without hypermethylation at the CpG_4.5, CpG_15, and CpG_17 sites of tumor tissue that underwent *PTGER4* methylation (*p* = 0.002 for RFS, 0.001 for PFS, and <0.001 for OS; Figure 2).

## 3. Discussion 

The TNM staging system is used to determine primary tumor size, lymph node status, and distant metastasis. It is a basic tool used to classify patients with CRC for prognosis prediction and treatment choice. The eighth edition of the AJCC Cancer Staging System is focused on the individual treatment and prognostic factors of CRC based on molecular biomarkers, such as *BRAF*, *KRAS*, and *NRAS* that are associated with chromosomal instability, mismatch repair, and microsatellite instability [17]. However, TNM classifiers cannot be used to control for heterogeneous characteristics within the same risk group [4]. The heterogeneity of the genetic features, epigenetic landscape, cell types, and tumor microenvironment corresponds to differences in the treatment response and clinical course among CRC patients [18]. Therefore, identifying robust genetic signatures that complement current TNM staging classification is crucial for prognosis because of high relapse rates [19]. 

The major epigenetic alterations observed in patients with CRC include DNA methylation, which influences genomic stability, gene transcription, and developmental processes in cells. Aberrant DNA methylation induces the development and progression of CRC [20]. During carcinogenesis, the hypermethylation of gene promoters mediated by DNA methyltransferases at CpG sites inhibits the expression of tumor-suppressor genes. Moreover, cancer germline genes and oncogenes are activated through the hypomethylation of gene promoters [21]. A systematic review revealed that the hypermethylation of gene promoters at CpG sites is an early event in the progression of CRC. Accordingly, highly sensitive and specific tools have been developed for the early screening, prognosis prediction, and recurrence prediction of CRC [22]. 

In the present study, we extracted tumor tissue and adjacent normal tissue from 208 patients with CRC and analyzed the influence of site-specific and tissue-specific methylation on disease progression. The hypermethylation of *PTGER4* promoter sequences at the CpG_4.5, CpG_15, and CpG_17 sites of primary tumor tissue corresponded to unfavorable 5-year RFS, PFS, and OS. In addition, *ZNF43* hypermethylation at CpG_5 in normal tissue was associated with a significantly poorer 5-year RFS and PFS relative to ZNF43 hypomethylation at CpG_5 in normal tissue. 

In this study, both *PTGER4* and *ZNF43* were classified as tumor-suppressor genes. The hypermethylation of *PTGER4* and *ZNF43* at certain loci enables molecular testing, which can be used in conjunction with TNM staging for a more effective prognosis of high-risk unfavorable CRC conditions.

The prostaglandin receptor encoded using *PTGER4* transmits upstream signals from the COX-2 enzyme and PGE2, thereby affecting tumorigenesis in the colon because PGE2 enhances various tumor-promoting factors, such as proliferation, expansion, angiogenesis, migration, and the invasion of colorectal carcinoma cells [23,24]. In addition, the level of the *PTGER4* receptor expression determines the growth and spread of tumor cells by activating ERK, PI3K/Akt, and cAMP/PKA/CREB signal transduction pathways and suppressing cAMP-dependent signal transduction [25,26]. The *PTGER4* receptor activated by PGE2 may induce anti-inflammatory effects along the cAMP/PKA axis and thus can benefit the treatment of CRC using data accumulated in vivo and in vitro [27]. The use of KAG-308, which is a PTGER4-selective agonist, reduces the risk of colorectal carcinogenesis by promoting mucosa healing and inhibiting macrophage activation [28]. *ZNF43* is a part of the C2H2-zinc finger (C2H2-ZNF) gene family [29], which is a C2H2-type ZNF-containing protein family and the largest transcription-regulatory factor family included in the human genome by rapid expansion through gene duplication [30]. A notable homologous sequence between family members exists both at the level of nucleotides and amino acids and enables the formation of a synchronous pattern and shared functions of gene expression. C2H2-ZNF proteins generally act as DNA-binding proteins, which mediate the recruitment of the transcriptional repressor KAP1/TRIM28 with a Krüppel-associated box (KRAB) domain [31]. Studies have investigated the pathophysiologic roles of KRAB-ZNF proteins that engage in biological events, such as genomic imprinting [32], cell differentiation [33], cell proliferation [34], apoptosis [35], and the cell cycle [36]. These properties of the KRAB-ZNF proteins have attracted attention in cancer studies. KRAB-ZNF proteins interact with KRAB-associated protein 1 (KAP-1), which serves as a transcriptional corepressor to mediate epigenetic alterations, such as DNA methylation and histone deacetylation in the promoter regions identified using the KRAB-ZNF proteins [37]. This promotes several biological processes, including DNA damage [38], epithelial–mesenchymal transition [39], the downregulation of p53 activity [40], and autophagy induction [41], which are crucial for carcinogenesis. According to Kel et al., *Z**NF43* hypermethylation is one of the DNA methylation biomarkers of CRC in the network that considers dynamic epigenetic changes [42]; this utility is indicated by the results of this study.

The hypermethylation of *PTGER4* at CpG_4.5, CpG_15, and CpG_17 corresponded to the undesirable prognosis of patients with CRC, as revealed by the analysis. The patients diagnosed with CRC were classified into two categories based on the hypermethylation status of candidate genes at several distinctive CpG sites. This classification was also adopted to investigate the histological differentiation and clinical prognosis in Australia [43]. Patients with CRC who underwent hypermethylation of *PDX1*, *EN2*, and *MSX1* at multiple CpG sites exhibited a high recurrence rate and a low survival rate, as demonstrated in a Korean study [44]. 

According to the results of this study, altered *ZNF43* methylation patterns in adjacent nontumorous tissue considerably affected people with CRC. This may be the result of field carcinogenesis, a theory that the accumulation of molecular and genetic alterations gradually change visually normal areas prior to the development of localized malignant tumors [45]. The distinction of transcriptomic signatures across histologically normal tissue in humans can help clinicians predict the possibility of cancer evolution and identify patients that require thorough examination after treatment [46]. Our results indicated that *ZNF43* hypermethylation in normal tissue adjacent to the lesion may recur and progress after surgery.

This study has several implications. The candidate genes were selected through a strictly three-step approach in terms of the gene expression in CRC, tissue-specific gene methylation, and related literature. For each gene, MS-PCR and EpiTYPER DNA methylation analyses were performed for qualitative and quantitative analyses, respectively. The high-resolution quantitative methylation profiling facilitated the high-sensitivity and high-specificity analysis of the methylation level of target genes at each CpG site for comparison between biopsy samples, which enabled the accurate identification of sites where notable epigenetic modifications occurred, thereby affecting the health conditions of patients with CRC. The study has some limitations. First, the Asian cohort was recruited from a single medical center in Taiwan and was in a small sample size with 208 CRC patients, limiting the generalizability of the results, which should be interpreted with caution. Large-scale and independent cohort studies can be conducted in the future to verify our findings. Second, the data regarding other risk factors related to CRC, such as dietary history, were not collected at that time. Moreover, the pieces of cancerous colorectal tissue and adjacent normal tissue were small, thereby increasing the chances of insignificant outcomes due to a sampling error. Finally, the expression level of each gene was not evaluated. Therefore, the influence of differential methylation on gene expression could not be determined in this study. 

## 4. Methods and Materials 

### 4.1. Study Design 

In this self-controlled cohort study, the methylation status and ratio of each CpG site were utilized to identify possible correlations between gene-related methylation and prognostic factors. Participants without correlated missing data were subjected to analyses of recurrence-free survival (RFS), progression-free survival (PFS), and overall survival (OS). To assess tissue-specific methylation, both tumor tissue and adjacent normal tissue with identical histopathology were considered.

### 4.2. Study Cohort and Specimens 

We collected the data of patients diagnosed with CRC who underwent surgical resection between 2006 and 2010 at Tri-Service General Hospital (TSGH), Taiwan. Ethical approval for this study was obtained from the TSGH Institutional Review Board (TSGHIRB; approval nos. 098-05-292 and 2-105-05-129). Informed consent was obtained from all subjects before the commencement of this project. The specimens included primary neoplasm tissue and nearby normal tissue extracted during surgery from patients with CRC. Inclusion criteria were individuals who were diagnosed with CRC at the Colon and Rectal Surgery Division of TSGH, needed surgical intervention, agreed with the study protocol, and signed the informed consent after surgeons’ explanation. Patients whose tumors, resected from surgery, were too small were excluded in this study. The tissues were stored at −80 °C before subsequent processing. The patients were required to regularly visit the outpatient department every 3 months for review in the first year after operation and once every 3 to 6 months subsequently over 5 years to enable prognostication. The cancer registration database of TSGH provided information on patient characteristics including sex, age at surgery, clinical staging, tumor location, tumor size, lymph node counts, histological grade, and adjuvant chemotherapy as well as follow-up data on recurrence, progression, and survival. 

The study involved 208 participants. For each individual, RFS, PFS, and OS were determined from the dates of operation to disease recurrence, tumor metas-tasis, death from any cause, or until the program finished on 31 December 2010. Figure 3 illustrates the flow of the study.

### 4.3. Gene Selection

Several studies on the surveillance of CRC have adopted the candidate gene approach, which accounts for the relationship between the genetic variation of selected genes and the phenotype of the disease [47]. The effects of gene expression on CRC prognosis were estimated using two databases: PREdiction of Clinical Outcomes from Genomic Profiles (https://precog.stanford.edu/ accessed on 10 February 2020) and Santa Cruz Genome Browser, University of California (https://genome.ucsc.edu/ accessed on 10 February 2020). The data were analyzed using the online tool Shiny Methylation Analysis Resource Tool (http://bioinfo-zs.com/smartapp/ accessed on 12 February 2020) to determine the divergent methylation of candidate genes in CRC neoplasms and normal tissue. Finally, studies on PubMed (https://pubmed.ncbi.nlm.nih.gov/ accessed on 15 February 2020) and Google Scholar (https://scholar.google.com.tw/ accessed on 15 February 2020) were reviewed. We selected *PTGER4* and *ZNF43* as the candidate genes due to the differential expression in the preliminary experimental results (Table 2) and limited resources. These two genes engaged in the signal pathways that are involved in cytokine inhibition, tumor cell proliferation, invasion, migration, metastasis, and carcinogenesis [48,49].

### 4.4. DNA Extraction and Bisulfite Modification

Genomic DNA samples were isolated from pathological tissue of CRC and adjacent normal tissue, and DNA bisulfite modification was performed according to the manufacturer’s instruction manual per a method described in a previous study [50].

### 4.5. Qualitative and Quantitative Analysis of DNA Methylation

Qualitative analysis was performed through a methylation-specific polymerase chain reaction (MS-PCR) to estimate the status of *PTGER4* and *ZNF43* gene methylation. The sequence of oligonucleotide primers, the annealing temperature of each primer used for amplification, and the final product sizes in MS-PCR are summarized in Table 6. The total volume of the reaction solution was 20 μL; the solution contained 9 μL of HotStart Taq Premix (RBC Bioscience, Taipei, Taiwan), 9 μL of pure water, 0.5 μL of 10 μM forward and reverse primers, and 1 μL of 10 μM bisulfite-modified DNA. The PCR cycling protocol reported in our previous study was adopted for MS-PCR [50].

Furthermore, we identified the CpG sites of *PTGER4* and *ZNF43* (Figure 1) and conducted a quantitative DNA analysis using the Agena Bioscience MassARRAY System with the EpiTYPER biochemistry kit (Agena Bioscience, San Diego, CA, USA) to detect the methylation level of each CpG site. The sequence of the oligonucleotide primers, the annealing temperature for amplification, and the product sizes for the EpiTyper assay are summarized in Table 6. The instructions followed for the PCR cycling and the workflow adopted for the EpiTYPER DNA methylation analysis have been elaborated on in our previous study [50]. 

### 4.6. Statistical Analysis

A Wilcoxon signed-rank test was used for a univariate analysis of the differences in the continuous variables between tumor tissue and adjacent normal tissue. For categorical variables, a Chi-square test and McNemar’s test were adopted. A univariate Cox proportional hazards regression model was established to evaluate the differences in the DNA methylation status between *PTGER4* and *ZNF43* with respect to RFS, PFS, and OS. Further, a multivariate Cox proportional hazards regression survival analysis was adopted to determine the correlations between gene methylation status and various prognostic factors with adjustments for various confounding factors, such as sex, age at surgery, clinical stage, lymph node counts, and histological grade. The influence of methylation levels at specific CpG sites on the 5-year RFS, PFS, and OS among patients with CRC was assessed through a Kaplan–Meier survival analysis; the differences between the survival curves were determined using a log-rank test. The statistical analyses in the study were conducted using IBM SPSS Statistics software (version 23). Statistical significance was indicated by a two-side *p* value < 0.05. To adjust for the alpha (type I error) inflation in this multiple-testing situation, a Bonferroni corrected two-sided *p* < 0.002, 0.003, and 0.004 (0.05 ÷ 12 to 17 different comparisons) was considered significant.

## 5. Conclusions

The current clinical staging system is based on the fundamental principle of risk stratification and treatment choice; however, the system lacks effectiveness because of clinically and genetically heterogeneous manifestations among patients with CRC. Many scientists and physicians have attempted to determine robust biomarkers to improve the effectiveness of prognosis. In this study, we identified and combined the loci of *PTGER4* and *ZNF43* DNA methylation, revealing their influence on recurrence, progression, and survival in CRC. Therefore, hypermethylation signatures can be used as a prognostic tissue marker that can assist clinicians in identifying patients with a high risk and to conduct suitable surveillance programs for them. 

## Figures and Tables

**Figure 1 ijms-23-10225-f001:**
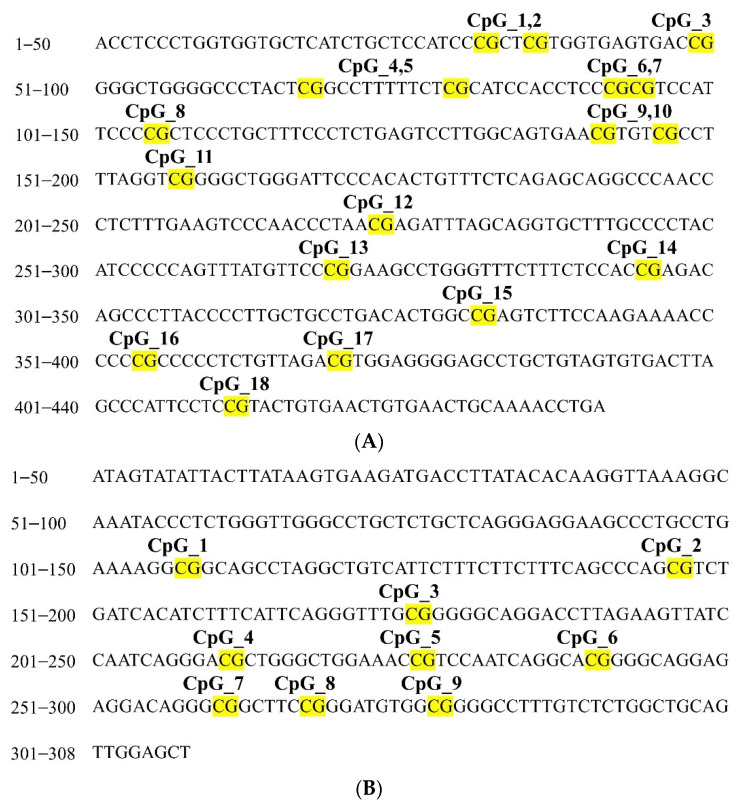
Schematic depiction showed the location of informative CpG sites in the (**A**) PTGER4 and (**B**) ZNF43 genomic loci. CG = CpG.

**Figure 2 ijms-23-10225-f002:**
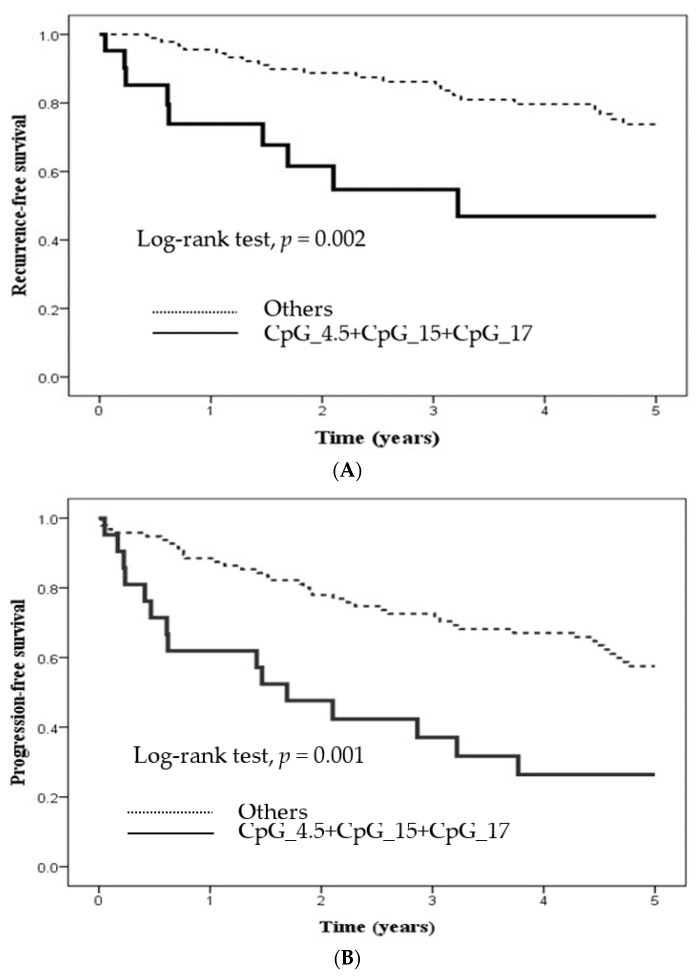
Clinical outcome of PTGER4 hypermethylation at CpG_4.5 + CpG_15 + CpG_17 from tumor tissue on 5-year (**A**) recurrence-free survival, (**B**) progression-free survival, and (**C**) overall survival of patients with colorectal cancer using Kaplan–Meier method.

**Figure 3 ijms-23-10225-f003:**
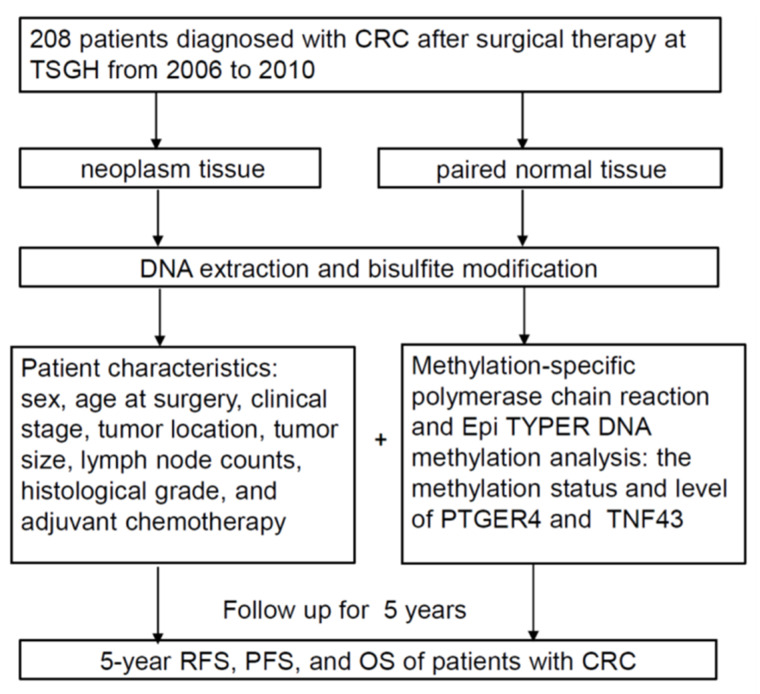
The study framework flow chart. CRC: colorectal cancer; TSGH: Tri-Service General Hospital; RFS: recurrence-free survival; PFS: progression-free survival; OS: overall survival.

**Table 1 ijms-23-10225-t001:** CRC patients’ characteristics and methylation status distribution (n = 208).

Variables	Total	Methylation Status			
		*PTGER4*		*ZNF43*	
		Normal	Tumor	Normal	Tumor
Sex					
Male	103 (49.5)	29 (42.0)	41 (59.4)	22 (31.9)	49 (71.0)
Female	105 (50.5)	36 (48.0)	47 (62.7)	20 (26.7)	57 (76.0)
χ^2^ (*p* value)		0.30 (0.581)	0.05 (0.820)	0.26 (0.614)	0.24 (0.625)
Age at surgery					
<50	35 (16.8)	12 (54.5)	12 (54.5)	5 (22.7)	15 (68.2)
≥50	173 (83.2)	53 (43.4)	76 (62.3)	37 (30.3)	91 (74.6)
χ^2^ (*p* value)		0.53 (0.465)	0.20 (0.654)	0.22 (0.640)	0.13 (0.715)
Stage					
I + II	106 (51.0)	33 (45.8)	42 (58.3)	29 (40.3)	53 (73.6)
III + IV	102 (49.0)	32 (44.4)	46 (63.9)	13 (18.1)	53 (73.6)
χ^2^ (*p* value)		<0.01 (1.000)	0.26 (0.608)	7.56 (0.006) ^a^	<0.01 (1.000)
Tumor location ^1^					
Colon	37 (20.1)	11 (40.7)	14 (51.9)	30 (28.3)	75 (70.8)
Rectum	147 (79.9)	50 (47.2)	66 (62.3)	7 (25.9)	23 (85.2)
χ^2^ (*p* value)		0.15 (0.702)	0.59 (0.443)	<0.01 (0.996)	1.63 (0.202)
Tumor size ^1^					
≤5 cm	114 (63.0)	40 (48.8)	46 (56.1)	18 (22.0)	57 (69.5)
>5 cm	67 (37.0)	20 (40.0)	33 (66.0)	19 (38.0)	40 (80.0)
χ^2^ (*p* value)		0.64 (0.422)	0.89 (0.346)	3.21(0.073)	1.26 (0.262)
Lymph node counts ^1^					
0–11	34 (18.4)	7 (35.0)	11 (55.0)	6 (30.0)	15 (75.0)
≥12	151 (81.6)	54 (47.4)	70 (61.4)	31 (27.2)	84 (73.7)
χ^2^ (*p* value)		0.61 (0.435)	0.09 (0.770)	<0.01 (1.000)	<0.01 (1.000)
Histological grade ^1^					
Well or Moderate	156 (89.7)	49 (44.5)	66 (60.0)	31 (28.2)	83 (75.5)
Poor or undifferentiated	18 (10.3)	8 (53.3)	8 (53.3)	5 (33.3)	9 (60.0)
χ^2^ (*p* value)		0.13 (0.715)	0.05 (0.831)	0.01 (0.913)	0.93 (0.336)
Adjuvant chemotherapy ^1^					
No	54 (29.3)	21 (52.5)	26 (65.0)	22 (23.7)	32 (80.0)
Yes	130 (70.7)	40 (43.0)	54 (58.1)	15 (37.5)	66 (71.0)
χ^2^ (*p* value)		0.67 (0.414)	0.31 (0.614)	2.03 (0.155)	0.76 (0.384)
5-year recurrence					
No	149 (71.6)	48 (45.3)	22 (57.9)	32 (31.1)	74 (71.8)
Yes	59 (28.4)	17 (44.7)	66 (62.3)	10 (24.4)	32 (78.0)
χ^2^ (*p* value)		<0.01 (1.000)	0.08 (0.779)	0.35 (0.554)	0.31 (0.580)
5-year progression					
No	116 (55.8)	36 (46.2)	45 (57.7)	28 (29.5)	68 (71.6)
Yes	92 (44.2)	29 (43.9)	43 (65.2)	14 (28.6)	38 (77.6)
χ^2^ (*p* value)		<0.01 (0.922)	0.55 (0.457)	<0.01 (1.000)	0.33 (0.568)
5-year all-cause death					
No	168 (80.8)	59 (49.6)	69 (58.0)	33 (27.7)	88 (73.9)
Yes	40 (19.2)	6 (24.0)	19 (76.0)	9 (36.0)	18 (72.0)
χ^2^ (*p* value)		4.48 (0.034) ^a^	2.12 (0.146)	0.34 (0.559)	<0.01 (1.000)

Abbreviations: CRC, colorectal cancer. ^1^ The sum of colorectal cancer cases does not match due to missing data; ^a^ *p* < 0.05.

**Table 2 ijms-23-10225-t002:** The methylation status distribution of candidate genes in tumor tissue and adjacent normal tissue from CRC patients.

Methylation Status			
*PTGER4* ^a^		*ZNF43* ^a^	
Normal	Tumor	Normal	Tumor
65 (45.1)	88 (61.1)	42 (29.2)	106 (73.6)

Abbreviations: CRC, colorectal cancer; ^a^ McNemar’s test (PTGER4, *p* = 0.017; ZNF43, *p* < 0.001).

**Table 3 ijms-23-10225-t003:** Comparison of methylation level at specific CpG sites of PTGER4 and ZNF43 in normal tissue and tumor tissue (n = 208).

	Normal			Tumor			*p* Value ^3^
	n ^1^	Median (Q1–Q3) (%)	Mean ± SD ^2^ (%)	n ^1^	Median (Q1–Q3) (%)	Mean ± SD ^2^ (%)	
*PTGER4*							
CpG_1.2	143	6 (3–8)	6.78 ± 2.4	144	6 (4.3–8)	7.37 ± 7.0	0.322
CpG_3	144	4 (3–6)	4.74 ± 2.7	140	3 (2–4)	4.64 ± 8.3	<0.001
CpG_4.5	141	11 (8–14)	12.06 ± 6.5	141	10 (7–13)	11.24 ± 7.5	0.044
CpG_6.7	142	6 (5–8)	6.65 ± 3.2	144	5 (3.3–7)	6.38 ± 7.4	<0.001
CpG_9.10	143	9 (8–12)	10.14 ± 4.4	143	7 (5–10)	8.71 ± 8.4	<0.001
CpG_11	144	10.5 (9–13)	11.56 ± 5.0	144	8 (6–11.8)	10.10 ± 8.5	<0.001
CpG_13	144	16 (13–18)	16.42 ± 5.8	144	11.5 (9–15)	13.07 ± 7.6	<0.001
CpG_15	142	16 (12–21)	18.05 ± 10.8	139	11 (6–16)	13.24 ± 11.1	<0.001
CpG_16	141	29 (26–33)	29.27 ± 5.8	143	23 (18–30)	25.36 ± 10.0	<0.001
CpG_17	84	18 (14–22)	18.21 ± 6.9	85	14 (10–18)	15.05 ± 8.2	0.052
CpG_18	143	27 (23–32)	28.31 ± 8.5	144	20 (16–29)	24.10 ± 13.8	<0.001
*ZNF43*							
CpG_2	143	4 (3.0–6.0)	5.17 ± 0.25	143	9 (4.0–29.0)	18.10 ± 1.56	<0.001
CpG_3	144	3 (2.0–3.0)	3.15 ± 0.26	144	9 (3.0–36.0)	20.39 ± 1.84	<0.001
CpG_4	131	1 (1.0–3.0)	1.98 ± 0.24	140	11 (2.0–36.0)	20.31 ± 1.87	<0.001
CpG_5	144	6 (4.0–9.0)	8.99 ± 0.77	143	16 (6.0–38.0)	23.08 ± 1.77	<0.001
CpG_6	84	3 (2.0–4.8)	4.07 ± 0.37	115	16 (4.0–39.0)	22.65 ± 1.82	<0.001
CpG_7.8	144	4 (4.0–5.0)	4.60 ± 0.22	144	9 (4.0–35.0)	21.24 ± 1.76	<0.001

Abbreviations: Q1, first quartile; Q3, third quartile; SD, standard deviation. ^1^ The sum of colorectal cancer cases does not match due to missing data; ^2^ indicates the ratio of DNA methylation; ^3^ the Bonferroni corrected values of *p* < 0.002 (0.05/17 different comparisons for each CpG or combine CpGs) were considered statistically significant.

**Table 4 ijms-23-10225-t004:** Cox regression analysis of methylation in certain CpG loci of PTGER4 and ZNF43 gene and 5-year clinical course.

	RFS		PFS		OS	
	cHR (95% CI)	aHR (95% CI) ^1^	cHR (95% CI)	aHR (95% CI) ^1^	cHR (95% CI)	aHR (95% CI) ^2^
*PTGER4* in tumor tissue						
all CpG sites						
hypomethylation	1.00 (Reference)	1.00 (Reference)	1.00 (Reference)	1.00 (Reference)	1.00 (Reference)	1.00 (Reference)
hypermethylation	1.22 (0.64–2.30)	1.03 (0.51–2.08)	1.47 (0.91–2.40)	1.33 (0.76–2.31)	2.18 (0.94–5.04)	1.77 (0.74–4.24)
*p* value	0.550	0.945	0.121	0.263	0.070	0.201
CpG_4.5 ^a^						
hypomethylation	1.00 (Reference)	1.00 (Reference)	1.00 (Reference)	1.00 (Reference)	1.00 (Reference)	1.00 (Reference)
hypermethylation	1.36 (0.71–2.58)	1.04 (0.51–2.12)	1.60 (0.97–2.63)	1.33 (0.76–2.30)	1.98 (0.85–4.58)	1.72 (0.72–4.09)
*p* value	0.354	0.917	0.064	0.317	0.112	0.222
CpG_15 ^a^						
hypomethylation	1.00 (Reference)	1.00 (Reference)	1.00 (Reference)	1.00 (Reference)	1.00 (Reference)	1.00 (Reference)
hypermethylation	1.86 (0.94–3.66)	1.06 (0.48–2.33)	1.39 (0.85–2.29)	1.07 (0.60–1.90)	2.11 (0.91–4.88)	1.55 (0.63–3.79)
*p* value	0.075	0.895	0.195	0.829	0.082	0.339
CpG_17 ^a^						
hypomethylation	1.00 (Reference)	1.00 (Reference)	1.00 (Reference)	1.00 (Reference)	1.00 (Reference)	1.00 (Reference)
hypermethylation	1.87 (0.79–4.41)	2.08 (0.79–5.48)	1.60 (0.84–3.04)	1.85 (0.89–3.84)	2.60 (0.83–8.17)	2.55 (0.80–8.05)
*p* value	0.155	0.137	0.156	0.099	0.103	0.112
*ZNF43* in normal tissue						
CpG_5						
hypomethylation	1.00 (Reference)	1.00 (Reference)	1.00 (Reference)	1.00 (Reference)	1.00 (Reference)	1.00 (Reference)
hypermethylation	1.81 (0.94–3.49)	2.33 (1.07–5.08) ^b^	1.78 (0.98–3.23)	2.42 (1.19–4.91) ^b^	1.16 (0.52–2.58)	1.13 (0.48–2.67)
*p* value	0.077	0.014	0.059	0.014	0.716	0.775

Abbreviations: RFS, recurrence-free survival; PFS, progression-free survival; OS, overall survival; cHR: crude hazard ratio; aHR: adjusted hazard ratio; CI: confidence interval. ^1^ Adjusted for sex, age at surgery, stage, lymph node counts, and histological grade; ^2^ adjusted for sex, age at surgery, stage, and lymph node counts; ^a^ missing values exist in the data; ^b^ *p* < 0.05 but did not approach the Bonferroni-corrected level of significance of *p* < 0.003 (0.05/15 different comparisons [3 outcome variables (RFS, PFS, and OS) with 5 CpGs or combined CpGs, resulting in 15 comparisons]).

**Table 5 ijms-23-10225-t005:** Impact of different combinations of CpG4.5, CpG15, and CpG17 of *PTGER4* hypermethylation in tumor tissue on 5-year prognosis.

	n ^1^	RFS		PFS		OS	
		cHR (95% CI)	aHR (95% CI) ^2^	cHR (95% CI)	aHR (95% CI) ^3^	cHR (95% CI)	aHR (95% CI) ^4^
CpG_4.5 + CpG_15 hypermethylation							
No	90	1.00 (Reference)	1.00 (Reference)	1.00 (Reference)	1.00 (Reference)	1.00 (Reference)	1.00 (Reference)
Yes	49	2.14 (1.12–4.07) ^a^	1.54 (0.74–3.18)	1.73 (1.06–2.84) ^a^	1.29 (0.73–2.28)	2.63 (1.19–5.80) ^a^	2.19 (0.95–5.05)
*p* value		0.021	0.248	0.030	0.385	0.016	0.066
CpG_4.5 + CpG_17 hypermethylation							
No	79	1.00 (Reference)	1.00 (Reference)	1.00 (Reference)	1.00 (Reference)	1.00 (Reference)	1.00 (Reference)
Yes	26	2.92 (1.37–6.22) ^a^	3.44 (1.53–7.72) ^a,b^	2.74 (1.53–4.92) ^a,b^	2.45 (1.26–4.76) ^a^	4.11 (1.62–10.4) ^a,b^	3.79 (1.46–9.88) ^a^
*p* value		0.005	0.003	0.001	0.008	0.003	0.006
CpG_15 + CpG_17 hypermethylation							
No	79	1.00 (Reference)	1.00 (Reference)	1.00 (Reference)	1.00 (Reference)	1.00 (Reference)	1.00 (Reference)
Yes	34	1.91 (0.89–4.11)	1.71 (0.73–4.01)	1.56 (0.87–2.78)	1.42 (0.74–2.74)	2.89 (1.17–7.11) ^a^	2.36 (0.93–5.97)
*p* value		0.099	0.214	0.133	0.295	0.021	0.071
CpG_4.5 + CpG_15 + CpG_17 hypermethylation							
No	96	1.00 (Reference)	1.00 (Reference)	1.00 (Reference)	1.00 (Reference)	1.00 (Reference)	1.00 (Reference)
Yes	21	3.18 (1.45–6.97) ^a,b^	3.26 (1.38–7.73) ^a^	2.77 (1.52–5.05) ^a,b^	2.35 (1.17–4.71) ^a^	4.79 (2.00–11.4) ^a,b^	4.32 (1.8–10.5) ^a,b^
*p* value		0.004	0.007	0.001	0.016	<0.001	0.001

Abbreviations: RFS, recurrence-free survival; PFS, progression-free survival; OS, overall survival; cHR: crude hazard ratio; aHR: adjusted hazard ratio; CI: confidence interval. ^1^ The sum of colorectal cancer cases does not match due to missing data; ^2^ adjusted for age at surgery, stage, lymph node counts, and histological grade; ^3^ adjusted for stage, lymph node counts, and histological grade; ^4^ adjusted for stage and lymph node counts; ^a^ *p* < 0.05; ^b^ the Bonferroni corrected values of *p* < 0.004 (0.05/12 different comparisons [3 outcome variables (RFS, PFS, and OS) with 4 combined CpGs, resulting in 12 comparisons]) were considered statistically significant.

**Table 6 ijms-23-10225-t006:** Primer sequences, annealing temperature, and product size for MS-PCR and EpiTYPER DNA methylation analysis of candidate genes.

Genes		Forward Primer (5′→3′)	Annealing Temperature (°C)	Product Size (bp)
*PTGER4*	M	F: GTTTTATTTCGTTCGTGGTGA	58.6	247
		R: AAAAAAAAAACCCAAACTTCC		
	U	F: GGGTTGGGGTTTTATTTGGTT	64.3	315
		R: CAACAAACTCCCCTCCACATC		
	Q	F: ATTTTTTTGGTGGTGTTTATTTGTT ^a^	57.9	440
		R: TCAAATTTTACAATTCACAATTCACA ^a^		
*ZNF43*	M	F: GGAGGAAGTTTTGTTTGAAAAGGC	61.3	323
		R: TTCTAAACTTCCGAAAAATCCTAAC		
	U	F: GAAGTTTTGTTTGAAAAGGTGG	59.5	328
		R: ACCATTTCTAAACTTCCAAAA		
	Q	F: AAGGTTAAAGGTAAATATTTTTTGGG ^a^	60.8	267
		R: TCCAACTACAACCAAAAACAAAAAC ^a^		

^a^: T7 tag: aggaagagag and cagtaatacgactcactatagggagaaggct were added as the T7 RNA polymerase promoter in the sequence of the forward primer and reverse primer, respectively. Abbreviations: MS-PCR, methylation-specific polymerase chain reaction; M, methylation; U, unmethylation; Q, quantitative analysis.

## Data Availability

No additional data are available.

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
