# Peer review of "Significance of Hypermethylation of Tumor-Suppressor Genes PTGER4 and ZNF43 at CpG Sites in the Prognosis of Colorectal Cancer"

_ijms, 2022, doi:10.3390/ijms231810225_

Round 1
Reviewer 1 Report
An interesting study assessing an important topic in CRC since the status of DNA methylation in primary tumor tissue and adjacent tumor-free tissue has been associated with the occurrence of aggressive disease and may aid personalized cancer treatments at early stages.
Some changes needed:
- an extensive linguistic revision required
- We believe this article is suitable for publication in the journal although major revisions are needed. The main strengths of this paper are that it addresses an interesting and very timely question and provides a clear answer, with some limitations. Certainly, the study is limited to an Asian population with a small sample size, and authors should further express this point.
- Second, the study included a widely varied patient population from an asian institute and the total number of patients analyzed was relatively small. Thus, the authors should better highlight the limitations of the current paper.
- The background of the changing scenario of medical treatment in CRC should be better discussed, and some recent papers regarding this topic should be included ( PMID: 35427471 ; PMID: 32684988 ).
Major changes are necessary.
Author Response
Review1
Comments and Suggestions for Authors
An interesting study assessing an important topic in CRC since the status of DNA methylation in primary tumor tissue and adjacent tumor-free tissue has been associated with the occurrence of aggressive disease and may aid personalized cancer treatments at early stages.
Some changes needed:
Responses: Thank you very much.
- an extensive linguistic revision required
Responses: We have done our best to revise our manuscript. Thank you very much.
- We believe this article is suitable for publication in the journal although major revisions are needed. The main strengths of this paper are that it addresses an interesting and very timely question and provides a clear answer, with some limitations. Certainly, the study is limited to an Asian population with a small sample size, and authors should further express this point.
Responses:
We agreed with the reviewer’s comments. We modified the limitation part in discussion that should be aware of. Please see the revised manuscript in Discussion section of the manuscript on page 14 line 376-379.
- Second, the study included a widely varied patient population from an asian institute and the total number of patients analyzed was relatively small. Thus, the authors should better highlight the limitations of the current paper.
Responses:
We agreed with the reviewer’s comments. We modified the limitation part in discussion that should be aware of. Please see the revised manuscript in Discussion section of the manuscript on page 14 line 376-379.
- The background of the changing scenario of medical treatment in CRC should be better discussed, and some recent papers regarding this topic should be included ( PMID: 35427471 ; PMID: 32684988 ).
Responses:
We were grateful for the reviewer’s advice and added recent medical literature to further discuss the development in CRC medical treatment. Please see the revised manuscript in Introduction section of the manuscript on page 2 line 56-65.
Major changes are necessary.
Responses: We have done our best to revise our manuscript. Thank you very much.
Reviewer 2 Report
The paper entitled Significance of Hypermethylation of Tumor-Suppressor Genes PTGER4 and ZNF43 at CpG Sites in the Prognosis of Colorectal Cancer is represented for peer review. Indeed, CRC is one of the most frequent invasive pathology detected so far. there is urgent need for reliable biomarkers to pecisely detect staging of CRC. Authors used PTGER4 as a major PGE2 receptor and ZNF4 as a member of KRAB-ZNF family. Study is well documented and extensively performed. Work is quite actual for cancer diagnostics and treatment.
I have good impression but several questions.
1. I haven t seen Approval from Ethics Comittee as well as number and date of Protocol.
2. Inclusion and exclusion criteria are not clarified. Please confine.
3. Sample is big enough but was it stratified by sex and age? Diet that also could influence methylation status of indicated genes is not specified.
4. Do you need to use Bonferroni correction for multiple testing in case of several CpGs? Please explain it.
5. I suggest additional validation by qPCR with specific primers to prove you data.
6. FIg.3 should be presented better scale up.
Author Response
Review2
Comments and Suggestions for Authors
The paper entitled Significance of Hypermethylation of Tumor-Suppressor Genes PTGER4 and ZNF43 at CpG Sites in the Prognosis of Colorectal Cancer is represented for peer review. Indeed, CRC is one of the most frequent invasive pathology detected so far. there is urgent need for reliable biomarkers to pecisely detect staging of CRC. Authors used PTGER4 as a major PGE2 receptor and ZNF4 as a member of KRAB-ZNF family. Study is well documented and extensively performed. Work is quite actual for cancer diagnostics and treatment.
I have good impression but several questions.
- I haven’t seen Approval from Ethics Comittee as well as number and date of Protocol.
Responses:
We valued the reviewer’s comments and modified sentences to record the Approval number from Ethics Committee. Please see the revised manuscript in Study Cohort and Specimens section of the Materials and Methods on page 3 line 102-104.
- Inclusion and exclusion criteria are not clarified. Please confine.
Responses:
We appreciated the reviewer’s comments. We added sentences to explain the inclusion and exclusion criteria. Please see the revised manuscript in Study Cohort and Specimens section of the Materials and Methods on page 3 line 107-110.
- Sample is big enough but was it stratified by sex and age? Diet that also could influence methylation status of indicated genes is not specified.
Responses:
We were grateful for the reviewer’s comments. Sample was not stratified by sex and age, but we conducted logistic regression analysis to adjust for sex and age in this study. Diet which could influence methylation status of indicated genes is an important issue. However, we conducted this retrospective cohort study and enrolled participants diagnosed in the Tri-Service General Hospital (TSGH), Taiwan, from 2006 to 2010. Data regarding registered patients—including sex, age at surgery, clinical stage, tumor location, size and histologic grade, lymph node counts, adjuvant chemotherapy, and survival—were collected from the TSGH's cancer registry database. It was certain that data on other factors associated with CRC risk such as diet were unavailable in the database. This is the limitation of this study. We added sentences to highlight this limitation. Please see the revised manuscript in Discussion section on page 14 line 380-381.
- Do you need to use Bonferroni correction for multiple testing in case of several CpGs? Please explain it.
Responses:
Thanks for your comments. The reviewer is correct. We were performed the Bonferroni correction for multiple testing in case of several CpGs. Bonferroni-corrected values of p<0.002 (0.05/17 different comparisons for each CpG or combine CpGs in Table 4), p<0.003 (0.05/15 different comparisons [3 outcome variables (RFS, PFS and OS) with 5 CpG or combine CpGs, resulting in 15 comparisons] in Table 5), and p<0.004 (0.05/12 different comparisons [3 outcome variables (RFS, PFS and OS) with 4 combine CpGs, resulting in 12 comparisons] in Table 6) were considered statistically significant in order to avoid a risk of the inflated Type I error. We added “To adjust for the alpha (type I error) inflation in this multiple-testing situation, a Bonferroni corrected 2-sided p<0.002, 0.003 and 0.004 (0.05÷12 to 17 different comparisons) was considered significant.” in the Statistical analysis section of this manuscript on page 6 line 191-193. We also added a footnote “The Bonferroni corrected values of p<0.002 (0.05/17 different comparisons for each CpG or combine CpGs), p<0.003 (0.05/15 different comparisons [3 outcome variables (RFS, PFS and OS) with 5 CpG or combine CpGs, resulting in 15 comparisons]), and p<0.004 (0.05/12 different comparisons [3 outcome variables (RFS, PFS and OS) with 4 combine CpGs, resulting in 12 comparisons]) were considered statistically significant.” In Table 4, Table 5 and Table 6, respectively.
- I suggest additional validation by qPCR with specific primers to prove your data.
Responses:
We were grateful for the reviewer’s advice. According to literature review (Stanssens et al., 2004; Ehrich et al., 2005; Yu et al., 2009; Bellido et al., 2010; Wu et al., 2016), Agena Bioscience MassARRAY system with DNA methylation analysis in our study was extremely accurate to assess quantitative methylation levels of multiple methylation sites, which was helpful to analyze more precisely. Although qPCR could be used to detect whether differential CpG sites methylation status could affect gene expression, we did not use the method of qPCR due to limited sources. In addition, we thought that we didn’t have fresh tissues to extract the total RNA and detect the gene expression by qPCR in this retrospective cohort study. So, the detection of gene expression from qPCR might not response the true gene expression in frozen tissues when DNA methylated, because of RNA was more degradation than DNA. This was one of our regrets. We added this situation in limitation. Please see the manuscript in Discussion on page 14 line 383-385.
References:
Bellido ML, Radpour R, Lapaire O, De Bie I, Hösli I, Bitzer J, Hmadcha A, Zhong XY, Holzgreve W. MALDI-TOF mass array analysis of RASSF1A and SERPINB5 methylation patterns in human placenta and plasma. Biol Reprod. 2010 Apr;82(4):745-50.
Ehrich M, Nelson MR, Stanssens P, Zabeau M, Liloglou T, Xinarianos G, Cantor CR, Field JK, van den Boom D. Quantitative high-throughput analysis of DNA methylation patterns by base-specific cleavage and mass spectrometry. Proc Natl Acad Sci U S A. 2005 Nov 1;102(44):15785-90.
Stanssens P, Zabeau M, Meersseman G, Remes G, Gansemans Y, Storm N, Hartmer R, Honisch C, Rodi CP, Böcker S, van den Boom D. High-throughput MALDI-TOF discovery of genomic sequence polymorphisms. Genome Res. 2004 Jan;14(1):126-33.
Wu Y, Li G, He D, Yang F, He G, He L, Zhang H, Deng Y, Fan M, Shen L, Zhou D, Zhang Z. Telomerase reverse transcriptase methylation predicts lymph node metastasis and prognosis in patients with gastric cancer. Onco Targets Ther. 2016 Jan 11;9:279-86.
Yu J, Tao Q, Cheng YY, Lee KY, Ng SS, Cheung KF, Tian L, Rha SY, Neumann U, Röcken C, Ebert MP, Chan FK, Sung JJ. Promoter methylation of the Wnt/beta-catenin signaling antagonist Dkk-3 is associated with poor survival in gastric cancer. Cancer. 2009 Jan 1;115(1):49-60.
- FIg.3 should be presented better scale up.
Responses:
We sincerely thank for reviewer’s comments. We have modified Fig.3 to be better presented. Please see the revised manuscript in FIg.3.
Reviewer 3 Report
In this manuscript, Chen et al. reported that epigenetic changes of tumor suppressor genes, PTGER4 and ZNF43 in patients with colorectal cancer (CRC). The methylation levels of PTGER4 and ZNF43 at certain CpG sites in CRC tissues correlated with poor prognostic factors such as overall survival, recurrence free survival and progression free survival. Authors clearly demonstrated the differential methylation status at specific loci of these tumor suppressor genes which plays a major role in identifying the prognosis of patients with CRC. The following minor concerns can further strengthen the quality of manuscript.
1. It is not clear that the rationale behind in picking up these 2 genes from the selected genes. It would be important to show the list of methylation status of selected genes.
2. Improve the quality of figure 3.

Author Response
Review3
Comments and Suggestions for Authors
In this manuscript, Chen et al. reported that epigenetic changes of tumor suppressor genes, PTGER4 and ZNF43 in patients with colorectal cancer (CRC). The methylation levels of PTGER4 and ZNF43 at certain CpG sites in CRC tissues correlated with poor prognostic factors such as overall survival, recurrence free survival and progression free survival. Authors clearly demonstrated the differential methylation status at specific loci of these tumor suppressor genes which plays a major role in identifying the prognosis of patients with CRC. The following minor concerns can further strengthen the quality of manuscript.
- It is not clear that the rationale behind in picking up these 2 genes from the selected genes. It would be important to show the list of methylation status of selected genes.
Responses:
We were grateful for the reviewer’s advice. We modified sentences to describe why we chose these two genes. Besides, we further added a table to show the list of methylation status of selected genes. Please see the revised manuscript in Gene Selection section of the Materials and Methods on page 4 line 135-145.
Table 1. The methylation status distribution of candidate genes in tumor tissue and adjacent normal tissue from CRC patients.
Methylation status |
|||
PTGER4a |
ZNF43a |
||
Normal |
Tumor |
Normal |
Tumor |
65(45.1) |
88(61.1) |
42(29.2) |
106(73.6) |
Abbreviations: CRC, Colorectal cancer; a McNemar's test (PTGER4, P = 0.017; ZNF43, P < 0.001)
- Improve the quality of figure 3.
Responses:
We sincerely thank for reviewer’s comments. We have modified figure 3 to improve the quality. Please see the revised manuscript in figure 3.
Round 2
Reviewer 1 Report
acceptance.
Reviewer 2 Report
Authors did great job and addressed all my issues. Thank you.